

# Seed traits and phylogeny explain plant distribution at large geographic scale

Kai Chen[1,2,5,6], Kevin S. Burgess[3], Fangliang He[4], Xiang-Yun Yang[1], Lian-Ming Gao[2], De-Zhu Li[1,2,5]

[1]Germplasm Bank of Wild Species in Southwest China, Kunming Institute of Botany, Chinese Academy of Sciences,
    Kunming, Yunnan 650201, China

[2]CAS Key Laboratory for Plant Diversity and Biogeography of East Asia, Kunming Institute of Botany, Chinese Academy of
    Sciences, Kunming, Yunnan 650201, China

[3]Department of Biology, College of Letters and Sciences, Columbus State University, University System of Georgia,
    Columbus, GA 31907-5645, USA

[4]Department of Renewable Resources, University of Alberta, Alberta, Canada

[5]Kunming College of Life Science, University of Chinese Academy of Sciences, Kunming, Yunnan 650201, China

[6]Key Laboratory of Insect Resources Conservation and Utilization in Western Yunnan, Baoshan University, Baoshan, Yunnan
    678000, China

*Correspondence to*: Lian-Ming Gao (gaolm@mail.kib.ac.cn), De-Zhu Li (dzl@mail.kib.ac.cn)

**Abstract.** Understanding the mechanisms that shape the geographic distribution of plant species is a central theme of biogeography. Although seed mass, seed dispersal mode and phylogeny have long been suspected to affect species distribution, the link between the sources of variation of these attributes and their joint effects to the distribution of seed plants remain poorly documented. This study aims to quantify the joint effects of key seed traits and phylogeny on the species' distribution. We collected seed mass and seed dispersal mode from 1,616 species of seed plants representing 554 genera of 130 families. We used 5,639,009 specimens to calculate species range size through ArcGIS10.2. Phylogenetic generalized least squares regression modeling and variation partitioning were performed to estimate the joint effects of seed mass, seed dispersal mode and phylogeny on species distribution. We found that species range size was constrained by seed dispersal mode and phylogeny. Seed mass and its intraspecific variation were also important in limiting species distribution, but their effects were different among species with different dispersal modes. Variation partitioning revealed that seed mass,

seed mass variability, seed dispersal mode and phylogeny together explained 40.44% of the variance in species range size.

Seed traits are not typically used to model the geographic distributions of seed plants. This study provides direct evidence

that seed mass, seed dispersal modes and phylogeny explain species distribution variation on a large geographic scale. Our

findings underscore the necessity to include seed traits and the phylogenetic history of species, together with existing

climate-based niche models, in predicting the response of plant geographic distribution to climate change.

**Keywords.** dispersal mode, geographic distribution, phylogeny, range size, seed mass, seed mass variability

## 1 Introduction

Understanding the ecological and evolutionary processes that govern the geographic range of species can provide insights

into their potential adaptive response to global climate change (Gaston and Fuller, 2009; Kubota et al., 2018). It is well

known that the geographic ranges of species can span 12 orders of magnitude, and even closely related species may vary

enormously in their geographic range (Brown et al., 1996). There are many factors contributing to this variation, although

dispersal ability and energy requirements associated with establishment and persistence in varying habitats are the two most

important factors (Morin and Chuine, 2006). Given that seeds are the predominately mobile stage of sessile plants and seed

mass generally reflects the amount of energy that a seed contains (Coomes and Grubb, 2003), it seems likely that seed mass

may play an important role in governing the geographic ranges of seed plants, although few studies have explored this

relationship (Morin and Chuine, 2006; Procheş et al., 2012).

Resource constraints on seed mass can influence the colonization and competitive ability of plant species (Chen et al.,

2018; Bu et al., 2019). For example, small-seeded species usually produce more seeds with small seed mass, and have a

greater potential to be dispersed by wind, and possess more opportunities to immigrate into new habitats or new climate

zones (Greene and Quesada, 2005; Morin and Chuine, 2006). Furthermore, seed mass has been shown to decrease along

increasing environmental extremes, indicative of the superior colonization ability of small-seeded species in low energy

habitats compared to that of large-seeded species (Procheş et al., 2012; DeMalach et al., 2019). Alternatively, large-seeded

species more often occupy habitats that have high levels of energy (i.e., tropical or low elevation habitats) and tend to be better competitors in these environments (Moles and Westoby, 2004), where they typically have higher germination rates (Galindez et al., 2009) and greater seedling survivorship (Mukherjee et al., 2019). However, the magnitude of seed mass

variation among species that contributes directly to limiting species range size remains unclear.

High levels of intraspecific variation for seed traits that influence seedling emergence and establishment may also be common (Cochrane et al., 2015). The response of intraspecific seed mass variation to heterogeneous environments may depend on plasticity genes or even the entire genome (Nicotra et al., 2010). Therefore, intraspecific seed mass variation reflecting a species' high genetic diversity can enable an adaptive response to varying environmental conditions (Yang et al.,

2016); high intraspecific seed mass variation may enable a species to occupy more habitats at a local scale (Silvertown, 1989; Sides and Sloat, 2014). Although intraspecific seed mass variation could be an important factor influencing the geographic distribution of plant species, few studies have evaluated this source of variation in a regional context.

The seed dispersal mode of a particular species, a key trait responsible for seed dispersal distance, can also greatly influence its geographic range (Chen et al., 2019b). The seed dispersal ability of a plant species is often a trade-off with other

life-history characteristics, such as seed persistence in the soil or the competitive ability of the emerging seedlings, which in turn can affect the survival, growth, and the reproduction of individuals (Nathan, 2001; Chen and Valone, 2017). However, we know little about the effects of dispersal modes on species distribution. Furthermore, because dispersal modes are often related to seed mass variation (Moles et al., 2007; Chen et al., 2019a), discerning the relative importance of seed mass and dispersal on the geographic distribution of seed plants is important but elusive.

Geographic distribution of species may also be phylogenetically correlated, because species of common descent that have the same time to dispersal, may co-occur and experience similar selection pressures in similar habitats, e.g., adaptive niche convergence (Losos, 2008; Grossenbacher et al., 2015). Alternatively, phylogenetic relationships can influence other ecological processes (e.g., niche partitioning in overlapping habitats) and variation in life-history traits, seed traits included, which in turn influence the distribution of species (Moles et al., 2005). Thus a species' age or degree of relatedness to other

species may invoke biogeographic limits to expansion (Martin and Husband, 2009) or promote the evolutionary divergence of species and the distribution and variation in seed traits (Donoghue et al., 2001; Moles et al., 2005). Although a species' range is likely dependent on its evolutionary history (Felsenstein, 1985), there are relatively few studies that have included phylogeny to better discern the effects of seed traits on species distribution.

    Given the above theoretical framework, we hypothesize that seed mass, intraspecific seed mass variation, dispersal

mode and phylogeny jointly influence species geographic range size. We predict that species possessing small seeds with high variability in seed mass and coupled with strong dispersal capacity, will have large range sizes and will be phylogenetically conserved. To determine the roles of seed traits and phylogeny on the distribution range of seed plants, we collected key seed traits (seed mass and seed dispersal mode) from 1,616 species of seed plants across the world, particularly the Mountains of Southwest China. We aim to address two questions to test our hypothesis: (1) Are there significant

phylogenetic signals associated with species range size? (2) What are the relative as well as joint effects of seed mass, seed dispersal and phylogeny on species range size?

## 2 Materials and methods

### 2.1 Seed mass data

Seed mass data (based on the weight of 1000 seeds per species) were obtained from the Germplasm Bank of Wild Species

(GBOWS: http://wwwgenobankorg/) and the Kew Gardens Seed Information Database (https://www.kew.org/kew-gardens). Our dataset contained 1,616 species from 19,991 populations, which represented 554 genera and 130 families of seed plants distributed globally (Fig.1). Seeds from two to 137 populations of each species (a total of 19,340 populations) were collected in China, and 651 populations of 138 species were from the Kew Gardens Seed Information Database. For each species, the mean 1000-seed weights of all populations were calculated. Seed mass variability (i.e., intraspecific seed mass variation),

ranging from zero to one, was calculated for each species as the difference between the minimum and the maximum values divided by the maximum value (Rozendaal et al., 2006).

## 2.2 Species range size

In this study, species range size was estimated through ArcGIS10.2 (Environmental Systems Research Institute Inc., Redlands, USA). Firstly, the specimen information of all the species in our study was obtained from the Global Biodiversity

Information Facility (GBIF.org,04 August 2019; GBIF Occurrence Downloadhttps://doi.org/10.15468/dl.umswqd), the Chinese Virtual Herbarium (http://wwwcvhorgcn) and the Biodiversity of the Hengduan Mountains and Adjacent Areas of South-Central China websites (BHMAASCC: http://hengduanhuhharvardedu/fieldnotes). Specimens lacking GPS location information, having duplication, containing incorrect coordinates, and those taken from gardens and small oceanic islands were filtered out of our analysis. In addition, species that were cultivated, introduced, invasive, or naturalized were also

excluded. After excluding these species, 5,639,009 specimens, representing 1616 seed plant species, were obtained (Fig.1). Secondly, *shapefiles* of specimen distributions were produced and transformed into $1° \times 1°$ (approximately12,388 $km^2$) grids through ArcGIS10.2. The range size of each species was calculated as the total number of occupied grids for the species.

## 2.3 Dispersal modes

Based on the published literature and floras, dispersal modes were classified into autochory (self-dispersal, e.g., by explosive

seed release from fruits or gravity), endozoochory (dispersal by animals through ingestion), exozoochory (dispersal by attachment to an animal body), and anemochory (dispersal by wind) according to the morphological features of their seeds or fruits (Pérez-Harguindeguy et al., 2013). For example, seeds or fruits with wings, hairs or pappus were considered wind diffused; seeds or fruits with hooks, spines or barbs were dispersed through exozoochory; seeds or fruits with anaril or flesh offering a succulent reward for consumers were classified as endozoochory; and seeds or fruits lacking modifications

pertaining to the other three categories were classed as autochory (unassisted dispersal) (Qi et al., 2014).

## 2.4 Construction of phylogenetic tree and statistical analyses

The phylogenetic tree was extracted from a previously published supertree (Qian and Jin, 2016) using the 'S.PhyloMaker' function in R package *phytools*, which was based on the APG classification of flowering plants (Zanne et al., 2014). The

'multi2d' function in the *ape* package was used to randomly resolve polytomies in the phylogenetic tree. To test the

phylogenetic signal in species distribution, 'phylosig' function in R package *phytools* was used to calculate Pagel's Lambda; this value ranges between 0 and 1. A value of 0 means that the evolution of the trait is independent of phylogeny, and a value of 1 indicates that trait evolution follows Brownian motion. Any value of Lambda significantly higher than zero can be regarded as a phylogenetic signal approaching Brownian motion to a different degree (Arène et al., 2017).

Because closely related species tend to have similar traits and interspecific analyses can be compromised by

phylogenetic correlation (Lynch, 1991), a phylogenetic generalized least squares (PGLS) regression was used to determine the partial effects of seed mass and seed mass variability on the range size of species for each dispersal mode, respectively. Collinearity inspection of the above models was based on a variance inflation factor (VIF); VIF larger than four suggests collinearity (Fox, 2002). In addition, the 'rda' function in *ade4* package was used to partition the variation of a species' range size explained by seed mass, seed mass variability, dispersal mode and genus (regarded as phylogeny). Finally, species range

sizes were also analyzed by Tukey's HSD tests to assess potential differences between dispersal modes. Before the analysis, the values of species range size were log10-transformed to meet the assumptions of normality. All statistical analyses in this study were conducted using R3.5.1 (R Core Team, 2018).

## 3 Results

### 3.1 Effects of phylogeny and dispersal modes on species' range size

The phylogenetic signal of species range size was significant (Lambda = 0.515, $P < 0.001$). The range sizes of phylogenetically related species were more similar than that for unrelated species (Fig.2). When compared to species dispersed by other modes, exozoochorous species had the largest mean distributional range size compared, while anemochorous species had the smallest mean distributional range size (Fig.3, Table A1).

### 3.2 Effects of seed mass and seed mass variability on species' range size


Based on the phylogenetic generalized least squares regression, seed mass was significantly and negatively correlated with

species distributional range size in exozoochorous (slope = -0.608, $P$ = 0.047; Fig.4c, Table A2) and anemochorous species

(slope = -0.458, $P$ < 0.001; Fig.4d, Table A2), but was not correlated for autochorous and endozoochorous species (Fig.4a

and Fig.4b, Table A2). Seed mass variability had a significantly positive relationship with species distributional range size in

autochorous (slope = 0.149, $P$ = 0.025; Fig.4a, Table A2), endozoochorous (slope = 0.371, $P$ < 0.001; Fig.4b, Table A2) and

anemochorous (slope = 0.456, $P$ < 0.001; Fig.4d, Table A2) species; no relation for exozoochorous species was detected

(Fig.4c, Table A2).

**3.3 Joint effects of key seed traits and phylogeny on species' distribution**

Variation partitioning showed that the collective effects of seed mass, seed mass variability, dispersal mode and phylogeny,

explained 40.44% of the variance associated with plant species distribution (Fig.5). Seed traits (including seed mass, seed

mass variability and dispersal mode) independently explained 15.70% of the variance in species distributional range size,

while phylogeny explained 32.73% of the variation associated with species distributional range size (Fig.5); 7.99% of the

explained variance was shared between seed traits and phylogeny (Fig.5).

**4 Discussion**

**4.1 The relationship between phylogeny and species distributional range size**

We found a significant phylogenetic signal associated with species range size (Fig.2). This result suggest that closely related

species have a more similar geographic distribution range than more distantly related species. Our study corroborates similar

results found in other studies (e.g., Hunt et al., 2005; Martin and Husband, 2009), but does not support those of Webb and

Gaston (2003), where the range size of closely related species were not more similar to each other than expected by chance

alone, and the correlation between species distributional range size and phylogeny was overestimated. This discrepancy may

be due to the evolutionary history of the various taxa involved as well as the heritability of their life-history traits, which can

play a critical role in the establishment and persistence of species, and thus their distributional range sizes (Angert and Schemske, 2005; Umaña et al., 2018). Seed traits associated with range size can also gradually change over evolutionary time, which can in turn increase the range of a species' distribution (Blomberg et al., 2003). Furthermore, the distribution range of a species can be influenced by its ecological tolerances associated with life-history traits (Geber and Griffen, 2003;

Canham et al., 2018), which can also have a heritable component. Our results imply that the geographic distribution of related species may have similar response patterns to climate change at the regional scale, due in part, to phylogenetic constraints on species distribution. Here, it seems likely that closely related species have evolved similar seed traits that result in shared adaptative strategies to climate change, although this causal mechanism requires further empirical study in the field.

**4.2 Effects of dispersal modes and seed mass on the distribution of species**

We found that exozoochorous species generally had larger distributional range sizes (Fig.3). This result is consistent with other studies (e.g., Dupré and Ehrlén, 2002) and can be explained by the longer dispersal distance of exozoochorous species compared to those having other modes of dispersal. Exozoochory is usually mediated by large mammals and birds, and has a greater dispersal distance than most other dispersal modes (Vittoz and Engler, 2007; Chen et al., 2019b). This finding

suggests that dispersal modes can have varying effects on the distributional range size of seed plant species.

We also found a significant relationship between seed mass and species range size in exozoochorous and anemochorous species after controlling for phylogeny (Fig.4, Table A2). Our results are, in part, distinct from previous studies that found a significant relationship between seed mass and species distributional range size (Morin and Chuine, 2006; Procheş et al., 2012), due perhaps, to differences in the taxonomic composition of our study compared to those previous. For example,

because our study is taxonomically broad, we were able to show that the effect of seed mass on seed dispersal distance may be different among species that have different dispersal modes. Based on our results for exozoochorous and anemochorous species, seed mass is a key factor for dispersal distance, but for autochorous and endozoochorous species, dispersal mode is more important than seed mass for dispersal distance. Given that small- and large-seeded species have been shown to adapt

to different habitats of heterogeneous environments (Silvertown, 1989), it seems likely that the autochorous and

endozoochorous species in our study may experience trade-offs between competition ability and dispersal ability through seed mass variation (Chen et al., 2018), which could result in a similar effect for seed mass on species distributional range size at large geographic scales.

We found significant relationships between intraspecific seed mass variability and species distributional range size (Fig.4). This result implies that species with a large amount of variation in seed mass have a greater adaptability to

heterogeneous environments and can occupy more sites. It is interesting to note that Sides and Sloat (2014) found that species with greater intraspecific variation in specific leaf area (SLA) have wider ecological breadth. Because of its potential role in modulating the responses of plant species to environmental changes, greater intraspecific functional variability may enable species to adjust to a wider range of competitive and abiotic conditions (Sides and Sloat, 2014; Manna et al., 2019). Furthermore, variability in seed mass optimizes plant functioning in a specific environment, and the requirements for optimal

seed mass functioning may differ in different environments (Rozendaal et al., 2006). Hence, our results suggest that plant species with a high intraspecific variation in seed mass have more potential adaptive strategies in highly heterogeneous environments.

Plastic responses of seed mass to heterogeneous environments may be related to molecular signals at a single gene or across the entire genome (Nicotra et al., 2010), which also has the potential to influence the distribution of species

(Savolainen et al., 2007). Distributional patterns of plant species may reflect the fact that individuals within a species have different levels of genetic variation that are also associated with seed mass, thus facilitating a species to adapt to a broad spectrum of environments (V öller et al., 2012). It is worth noting that the regression slopes of seed mass variability were statistically different among dispersal modes in our study. This could be explained by different seed mass allocation strategies among dispersal modes (Chen et al., 2018; Chen et al., 2019a).

**4.3 Joint effects of seed mass, seed dispersal and phylogeny on species distributional range size**

Seed traits and phylogeny jointly affect species distribution in our study, indicating that species distribution is limited by

ecological and evolutionary determinants (Fig.5). There are two possible reasons for this relationship. Firstly, both the evolution of seed mass and dispersal mode are phylogenetically constrained (Gallagher and Leishman, 2012; Chen et al., 2018). Secondly, seed mass and seed dispersal mode are not evolutionarily independent, i.e., evolutionary divergences in dispersal syndrome explain divergences in seed mass (Moles et al., 2005). In addition, more than 50% of the variance associated with species distribution in our study remains unexplained, which may imply that climatic tolerance, competition and colonization ability are also important factors for species distribution (Morin and Chuine, 2006; Lyon et al., 2019).

## 5 Conclusions

This study provides evidence that seed mass, intraspecific seed mass variation, seed dispersal modes and phylogeny explain species distribution variation on a large geographic scale. Our most striking results are, firstly, that species distributional range size was constrained by phylogeny, seed mass and its intraspecific variability, and seed dispersal mode. Secondly, seed mass variability was much more important than seed mass for species distribution; the effects of seed mass and seed mass variability on species distribution varied among dispersal modes. Thirdly, variation partitioning revealed that seed mass, dispersal mode and phylogeny together explained over 40% of the variance associated with species distribution. To provide a deeper understanding of the variation associated with species distribution, we suggest future studies include more functional seed traits (such as germination rate, germination time, seed lipid or polysaccharide content, fruit type), as well as experimental field trials that test the ecological and evolutionary role of seed mass variation on the establishment and colonization of seed plants.

*Data availability.* The data are available from the freely accessible databases cited in the manuscript.

*Authors contribution.* DZL, LMG and FH designed the study; KC and XYY collected data; KC conducted statistical analysis and generated the graphs; KC, KSB and LMG wrote the manuscript; DZL, FH and XYY revised the manuscript. All authors reviewed and approved the final manuscript.



*Competing interests.* All authors have no conflict of interest.

*Acknowledgements.* We are grateful to Xie He, Jie Cai, Ting Zhang, Jian-Jun Jin, Hua-Jie He, Tuo-Jing Li and other supporting staff of the Germplasm Bank of Wild Species for assistance in seed mass data or specimen data collection. We also thank Ming-Cheng Wang for helping calculate species range size.

*Financial support.* This study was supported by the Strategic Priority Research Program of Chinese Academy of Sciences (XDB31000000), the National Key Basic Research Program of China (2014CB954100), and the Program of Science and

Technology Talents Training of Yunnan Province (2017HA014).

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



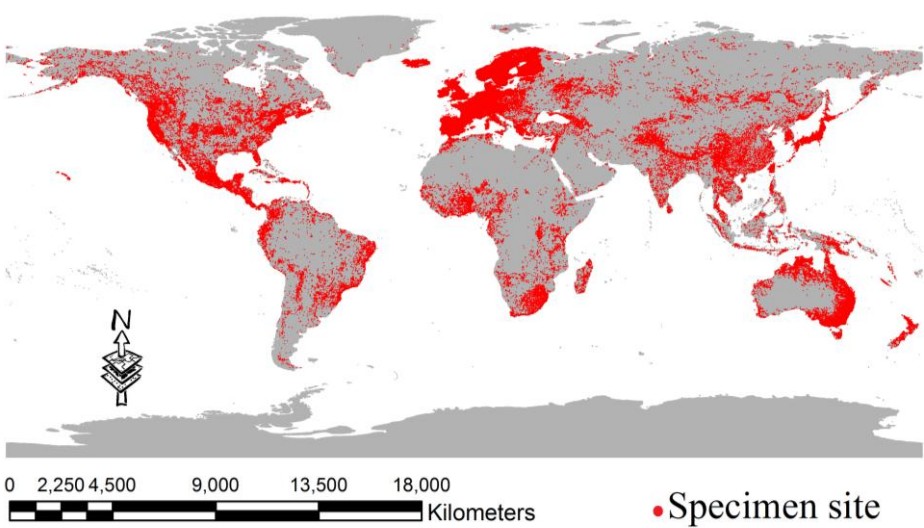


**Figure 1. The Global distributions of 5,639,009 plant specimen records used in this study.**



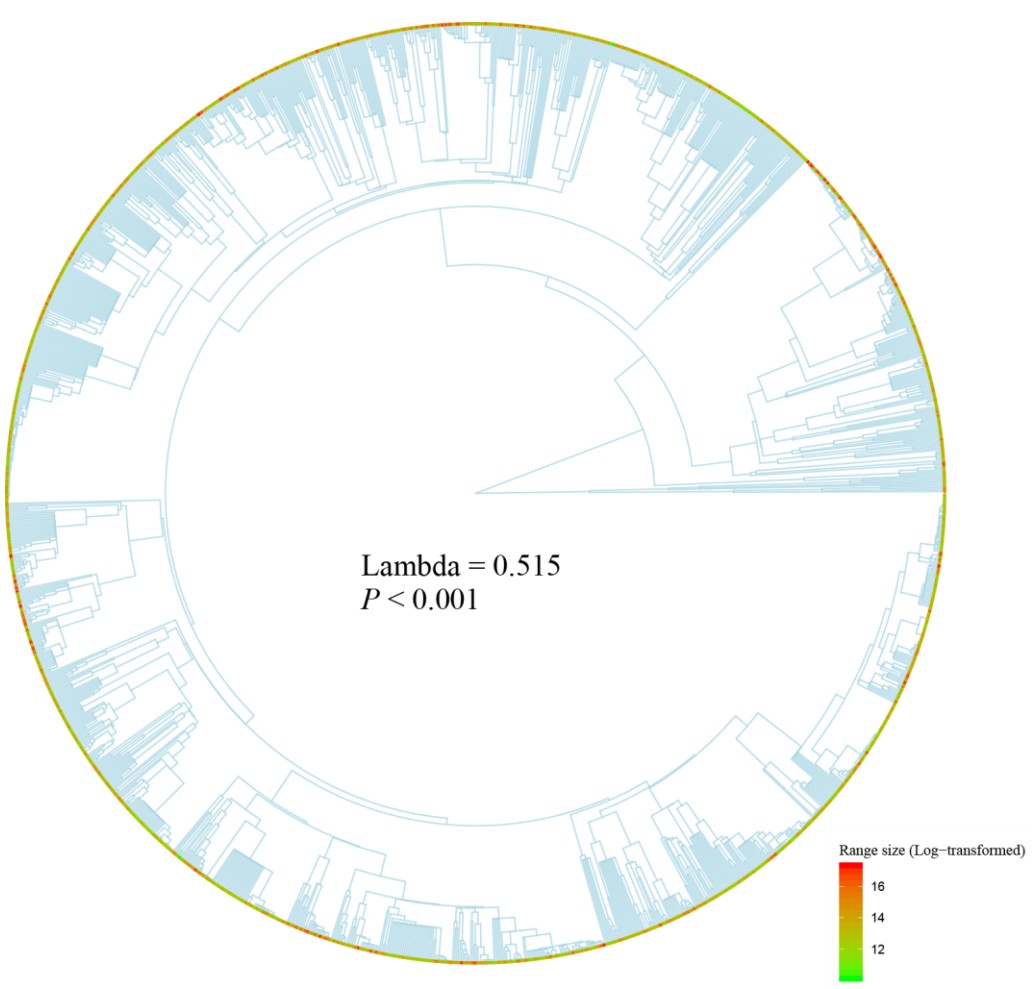

**Figure 2. Phylogenetic signal of 1,616 species of seed plants for distributional range size.**



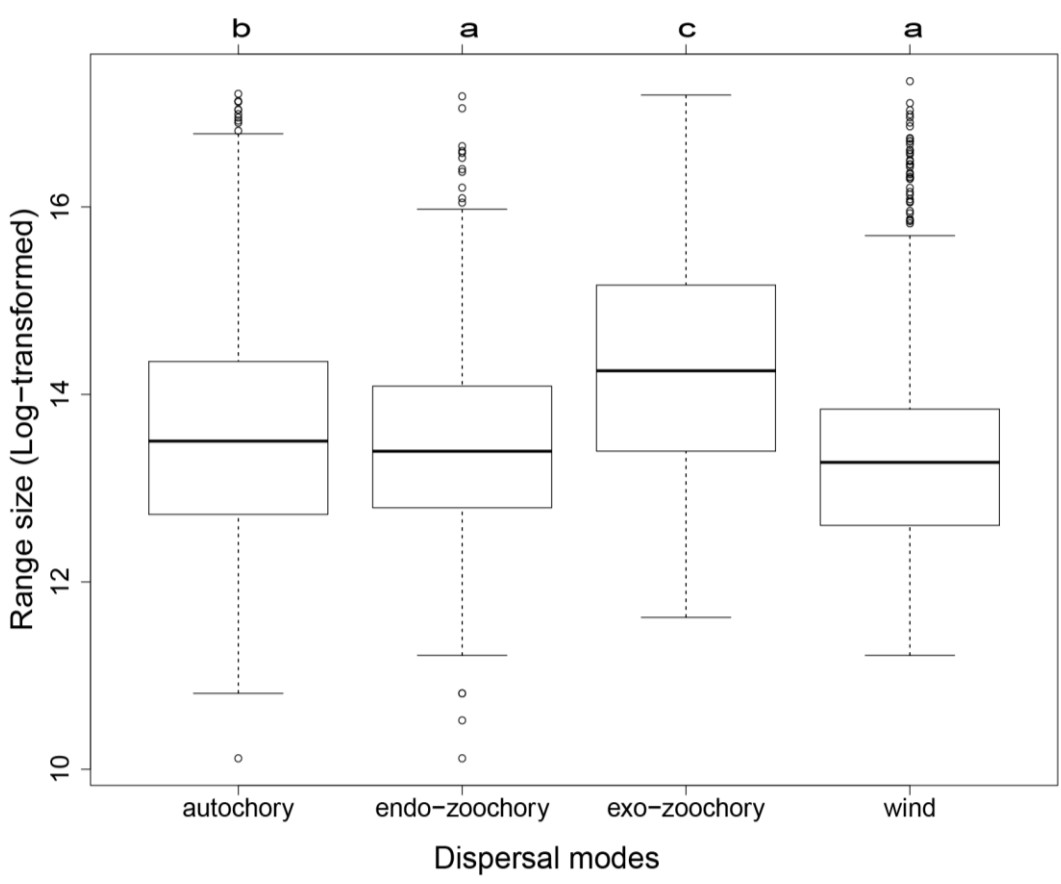

**Figure 3. Variation in species distributional range size for four dispersal modes. Different letters indicate statistically significant differences as tested by Tukey's HSD analysis.**

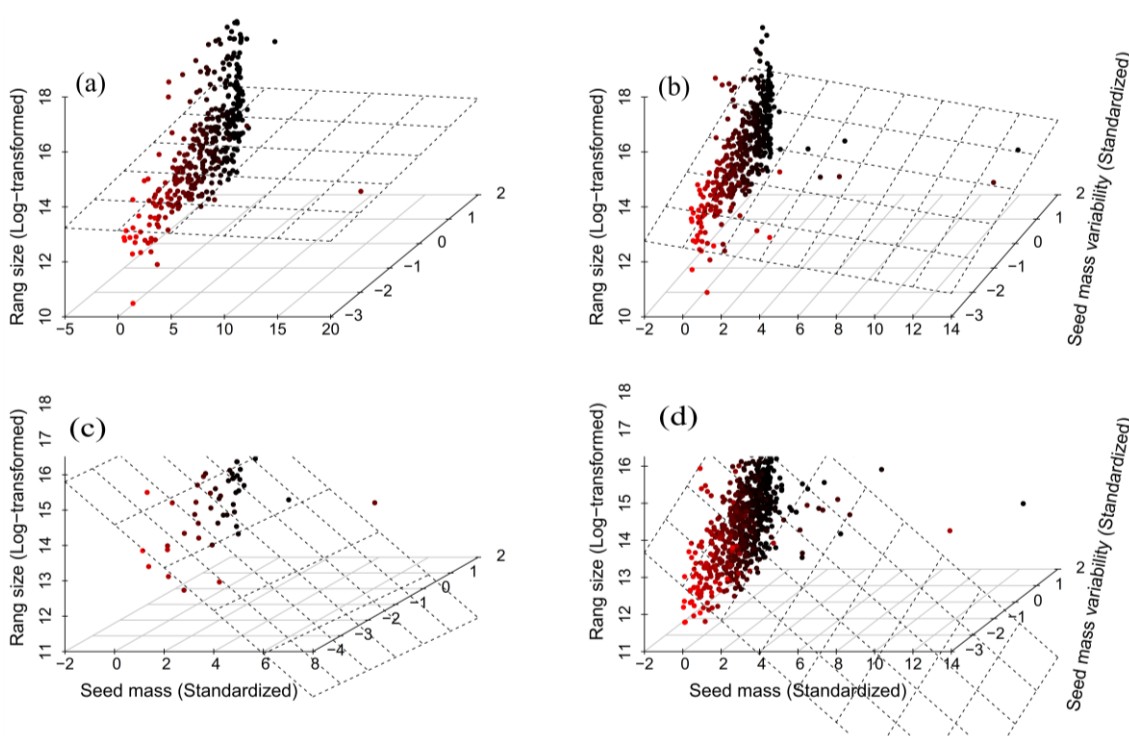

**Figure 4. Effects of seed mass and seed mass variability on species range size in autochorous (a), endozoochorous (b), exozoochorous (c) and anemochorous (d) species. Colors represent different values of *y* (i.e., rang size).**





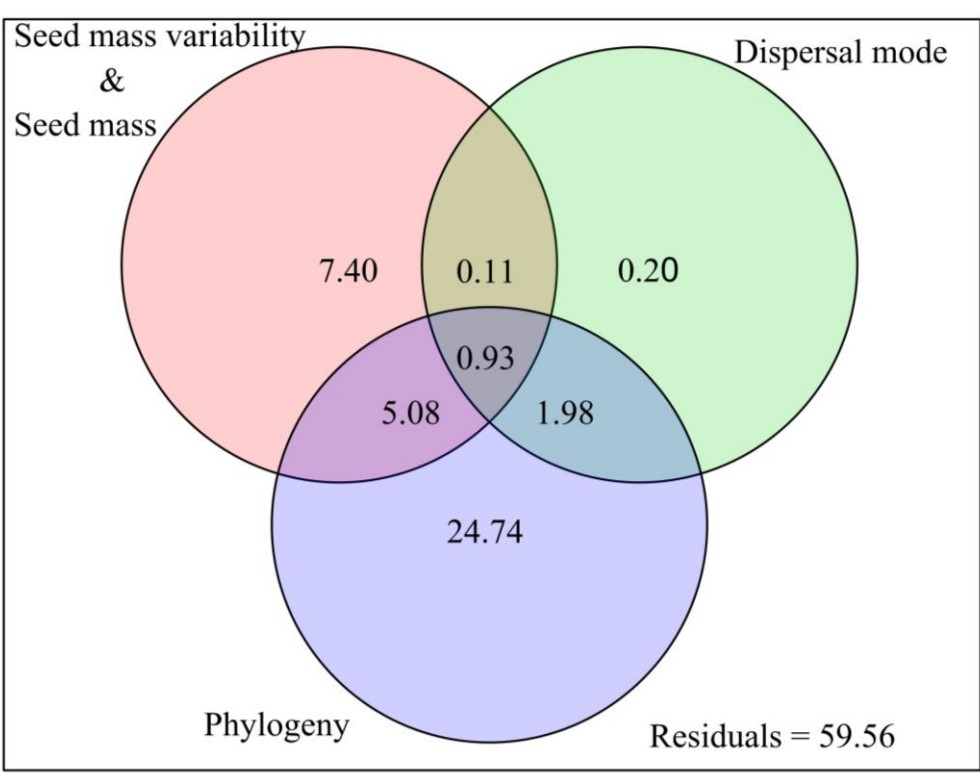


**Figure 5. Variation partition of seed mass, seed mass variability, dispersal mode and phylogeny for species distributional range size.**





## APPENDICES

**Table A1. Comparison among dispersal modes for linear hypotheses of mean species distributional range size.**

| Linear hypotheses for range size | Estimate ±SE | t-value | *P*-value |
|---|---|---|---|
| Endozoochory- Autochory == 0 | -0.213 ±0.081 | -2.626 | 0.040 |
| Exozoochory - Autochory == 0 | 0.668 ±0.144 | 4.637 | <0.001 |
| Wind - Autochory == 0 | -0.288 ±0.073 | -3.891 | <0.001 |
| Exozoochory - Endozoochory == 0 | 0.881 ±0.142 | 6.213 | <0.001 |
| Wind - Endozoochory == 0 | -0.075 ±0.070 | -1.077 | 0.691 |
| Wind - Exozoochory == 0 | -0.956 ±0.138 | -6.934 | <0.001 |






**Table A2. Summary of the phylogenetic generalized least squares used to describe log-transformed species distributional range size.**

| Dispersal mode | Independent variables | Estimate±SE | t-value | *P*-value | VIF |
|---|---|---|---|---|---|
| Autochory | Seed mass | -0.019±0.228 | -0.083 | 0.933 | 1.001 |
| | Seed mass variability | 0.149±0.066 | 2.251 | 0.025 | 1.001 |
| Endozoochory | Seed mass | -0.119±0.120 | -0.991 | 0.323 | 1.003 |
| | Seed mass variability | 0.371±0.046 | 8.141 | < 0.001 | 1.003 |
| Exozoochory | Seed mass | -0.608±0.300 | -2.023 | 0.047 | 1.036 |
| | Seed mass variability | -0.073±0.157 | -0.465 | 0.643 | 1.036 |
| Wind | Seed mass | -0.458±0.108 | -4.220 | < 0.001 | 1.002 |
| | Seed mass variability | 0.456±0.037 | 12.272 | < 0.001 | 1.002 |