# Peer review of "Seed traits and phylogeny explain plant distribution at large geographic scale"

_Biogeosciences, 2020_

## Referee Comment (RC1) · Anonymous Referee #1 · 3 Jul 2020

Chen et al.' MS entitled "Seed traits and phylogeny explain plant distribution at large geographic scale" addresses how seed mass, seed dispersal mode and phylogeny jointly affect species distribution based on a fascinating dataset of seed traits from 1616 seed plant species mainly from China. The authors quantify the effects of seed mass, seed dispersal mode and phylogeny on species distribution, and observe that species range size is constrained by seed dispersal mode and phylogeny. The results of the study provide the direct evidence that seed mass, seed dispersal modes and phylogeny explain species distribution variation on a large geographic scale, which is very important for understanding the seed plant biogeographic pattern and predicting the response of plant geographic distribution to climate change.

In general, the text is well written, although some clarifications are needed in some

places. The language is appropriate. The discussion and conclusions seem to be fairly well supported by the results. However, I have a few minor concerns about the methods and results on improving the manuscript. Once these concerns are resolved I think the manuscript would potentially deserve to be published.

The presentation of the 1000 seeds per species of seed mass in the Materials and methods section is not clear. How the 1000 seeds per species were selected? Did all the species have at least 1000 seeds used for measurement? Please clarify.

The authors used records of species specimens to quantify species distribution range size, while some other related studies adopt SDM models to estimate species distribution. Why do the authors use specimen records to quantify species distribution range size rather than using SDM models? Which method is more appropriate? Why?

The authors mentioned that they tested the models based on a variance inflation factor (VIF) in the methods (L122). It is remarkable that the VIF values are same for different predictors (see Table A2), which needs to be clarified in the results and discussion respectively.

Specific comments:

P7 L132: "by other models" what kind of "other models" need to be clarified. P9 L180: in our study can be deleted. P10 L194: the distribution of species should be the distributional range size of species. P10 L201: Change "Seed traits and phylogeny jointly affect species distribution in our study" to "Our results here demonstrated that seed traits and phylogeny jointly affect the species distribution, ..." P10 L201: Seed traits and phylogeny jointly affect species distribution In Figure 2, Lambda-value and P-value need to be clarified in the caption.

---

## Author Comment (AC1) · 13 Jul 2020

The presentation of the 1000 seeds per species of seed mass in the Materials and methods section is not clear. How the 1000 seeds per species were selected? Did all the species have at least 1000 seeds used for measurement? Please clarify. Authors Response: Thank you for your questions. Mature seeds were collected at the start of natural dispersal and dried in a drying room where the relative humidity and temperature were maintained at 15% and 15°C, respectively. Moisture was drawn out of the seeds until water content was the same as that in the air. After drying, 250 seeds were randomly selected to be measured to the nearest 0.1 mg, and repeated four times. In order to protect germplasm resources, each species has more than 1,000 seeds in Germplasm Bank of Wild Species in Southwest China. We will clarify the 1000-seed

weight in the Method.

The authors used records of species specimens to quantify species distribution range size, while some other related studies adopt SDM models to estimate species distribution. Why do the authors use specimen records to quantify species distribution range size rather than using SDM models? Which method is more appropriate? Why? Authors Response: Thank you for your questions. Distribution range size calculated through SDM models are under "ideal" conditions which dispersal ability, time and other important plant traits are not considered as limiting factors. Therefore, this range size is also called potential distribution range size. Usually some potential distribution ranges are not occupied by species, due to the constraint of dispersal ability. While, distribution range determined by species specimens is the species "actual" distribution. Especially, range size estimating on a large number of specimens should be the unbiased estimation of species distribution range size. Using records of species specimens to quantify species distribution range size is more appropriate when having massive amount of specimen data.

The authors mentioned that they tested the models based on a variance inflation factor (VIF) in the methods (L122). It is remarkable that the VIF values are same for different predictors (see Table A2), which needs to be clarified in the results and discussion respectively. Authors Response: Thanks a lot for pointing it. The VIF (variance inflation factor) of the i-th variable is defined as: $VIF_i = 1/(1 - R_i^2)$, where $R_i^2$ is the goodness-of-fit of the linear model for $x_i$ based on all other predictive variables. Here, we have only two predictive variables and the $R^2$ values of the two variables are equal. We will clarify in the caption of Table A2.

Specific comments: P7 L132: "by other models" what kind of "other models" need to be clarified. Authors Response: Here, "other models" represented autochory, endozoochory and anemochory. We reworded the sentence to make it clear in the revised manuscript.

P9 L180:in our study can be deleted. Authors Response: Have done.

P10 L194: the distribution of species should be the distributional range size of species. Authors Response: We have changed accordingly.

P10 L201: Change "Seed traits and phylogeny jointly affect species distribution in our study" to "Our results here demonstrated that seed traits and phylogeny jointly affect the species distribution, ..." Authors Response: We have reworded the sentence accordingly.

In Figure 2, Lambda-value and P-valueneed to be clarified in the caption. Authors Response: Lambda-value and P-value have been clarified in the caption as "P< 0.001 means that the phylogenetic signal of range size is significant, and Lambda-value = 0.515 implies that the evolution model of species range size is different from Brownian motion."

Please also note the supplement to this comment:
https://www.biogeosciences-discuss.net/bg-2020-186/bg-2020-186-AC1-supplement.pdf

---

## Referee Comment (RC2) · Anonymous Referee #1 · 16 Jul 2020

Many thank sfor the responses.

---

## Referee Comment (RC3) · Anonymous Referee #2 · 8 Aug 2020

The authors focus on an interesting topic (how seed mass affects species range size). The Introduction is generally well written (can be improved by presenting clear predictions and I have a couple of minor comments below). However, the Methods section is so poorly written, with many flaws and errors. I got disappointed after reading the Methods section, and therefore cannot trust the Results section and therefore skipped the Discussion section (since I cannot judge whether the results are justified). The authors seemingly present many fancy analyses but are not really clear why to use them. Luckily, I know clearly all these methods and present my suggestions here.

First, please minimise errors, otherwise, audience cannot understand you. The URL of GBOWS in L85 cannot be opened (I guess you are saying "http://www.genobank.org/"?). L112 is wrong – "S.PhyloMaker" is not implemented in

the phytools package. They are separate things. Also, you should use the updated version, known as "V.PhyloMaker" (Jin and Qian 2019. Ecography Vol. 42). The most severe error is in L123 – there is no 'rda' function in ade4 package. This error makes me unable to examine whether your variation partition (the most important part of this study) is suitable or not, because the method is unexposed.

Second, please upload your data to make the study reproducible and meanwhile give clear descriptions. L86 – how did you choose these 1616 species from these two databases? This should be an important piece of information in this whole study, otherwise, it may produce biases. Will the species in China be overrepresented and cause potential biases to the analyses? Have you checked and standardised nomenclatures of species and how? L88 – two to 137 populations per species in China. Then, how many populations per species from the Kew SID? Another important issue is that how did you treat species with more than one type of morphological adaption (L104-110)? Many species can be dispersed by several modes, for example, by both endozoochory and exozoochory. What about species that are dispersed by water or ants? Please give a sample size of species in each category. I suggest uploading the compiled data you used in the analysis, since they are from two databases anyway.

Third, you mentioned many important issues in analysing data, but you stop at only a brief mention. You need to continue with why they are important issues and how you have solved them. That is, you need to explain what this analysis does and why it is relevant to your data, rather than just referring to an R function. Since I could not find the function you mentioned in doing variation partitioning (L123), I doubt whether it can really be suitable to your data structure where you have a continuous variable (seed mass), categorical variables (dispersal mode, genus) and proportion data (seed mass variability) all mixed? In L122-123, it is good that you realise multicollinearity, but how did you treat variables with high collinearity issue? L126 – the "assumptions of normality" refers to the residuals, but have you checked your model residuals? L89-91: Why did you use this index to show seed mass variability? Why not use (for example)

coefficient of variation, which is more commonly used?

Fourth, there are some analytical flaws. L101 – 1 degree × 1 degree grids cannot be of the same size across the globe. This is a wrong procedure in calculating species range size, because it seems you did not consider projection. Since your grids have different sizes near the equator and at high latitudes, how could the species range size be comparable across species? In addition, I doubt the use of genus to surrogate phylogeny. In the variation partitioning, it seems you did not incorporate the phylogenetic correlation among species, which violates your previous sentence "closely related species tend to have similar traits and interspecific analyses can be compromised by phylogenetic correlation". In L125 – are you saying you did ANOVA with post-hoc Tukey HSD tests? What package did you use to do the tests? Again, you need to take phylogeny into account, otherwise, residuals do not fit model assumptions.

Minor comments:

The Introduction is generally written but can be improved with clear predictions. You only present study goals and predictions until the last paragraph (L74-75 is not a clear hypothesis). Why you make such hypotheses and what makes your study novel are essential throughout the whole Introduction section. For example, in L39-40 "few studies have explored this relationship", then what have they found? What makes your study different from these previous ones? L50 – what do you plan to do about this gap?

L66: Rephrase "have the same time to dispersal". No idea what this means – this line either has a grammar error or is delivered wrongly.

All references do not have years – how could they be matched with citations in the main text?

Figure 1 and Figure 2 actually tell little information. I suggest removing Figure 2 since no tip can be seen with these many species. For Figure 1, I suggest using grids to show numbers of specimen records (same unit as the grids used in range size), which

can avoid overlapping.

Figure 4: Is range size log-transformed (Figure 4 and Table A2) or log10-transformed (the main text)? What is the flat panel in each figure? Here, you standardised predictors, but this information is not given in the Methods section.

Figure 5: Seed mass and seed mass variability are two separate variables, but why are they combined in the variation partition?

The results in Figure 3 and Figure 5 are not reliable, due to the flawed methods.

---

## Author Comment (AC2) · 17 Aug 2020

Authors response to referee comments

First, please minimise errors, otherwise, audience cannot understand you. The URL of GBOWS in L85 cannot be opened (I guess you are saying "http://www.genobank.org/"?). Response: Thank you. GBOWS is the abbreviation of the Germplasm Bank of Wild Species in Southwest China. The link has now been updated in the revised manuscript. (http://www.genobank.org/)

L112 is wrong – "S.PhyloMaker" is not implemented in the phytools package. They are separate things. Also, you should use the updated version, known as "V.PhyloMaker" (Jin and Qian 2019. Ecography Vol. 42). Response: Thank you. Indeed,

"S.PhyloMaker" is not implemented in any R packages. We will use "V.PhyloMaker" in the revised manuscript if we get a chance to resubmit our manuscript.

The most severe error is in L123 – there is no 'rda' function in ade4 package. This error makes me unable to examine whether your variation partition (the most important part of this study) is suitable or not, because the method is unexposed. Response: Thanks for pointing this out. The 'rda' function is in the vegan package. Change will be made in the revised manuscript.

Second, please upload your data to make the study reproducible and meanwhile give clear descriptions. Response: Thank you. We have uploaded the R code of the statistical models and our data will also be uploaded to the Dryad Digital Repository upon acceptance.

L86 – how did you choose these 1616 species from these two databases? This should be an important piece of information in this whole study, otherwise, it may produce biases. Will the species in China be overrepresented and cause potential biases to the analyses? Have you checked and standardised nomenclatures of species and how? Response: Our analysis was mainly based on the collection a ten-year seed conservation initiative in China, i.e. the Germplasm Bank of Wild Species in Southwest China. We didn't preferably select one group of species over another but instead we used the existing data set available by the end of Aug., 2018 by that time I completed my PhD thesis. All the 1616 species are distributed in China, and about 30% of these species are endemic to China. For all the species used in our analysis, the scientific names were checked and standardized according to the Plant List (http://www.theplantlist.org/), which is a working list of all known plant species.

L88 – two to 137 populations per species in China. Then, how many populations per species from the Kew SID? Another important issue is that how did you treat species with more than one type of morphological adaption (L104-110)? Response: One to 6 populations per species were from the Kew SID. The species list of the Kew SID is

also included in the Germplasm Bank of Wild Species in Southwest China. Different varieties and subspecies were considered as the same species.

Many species can be dispersed by several modes, for example, by both endozoochory and exozoochory. What about species that are dispersed by water or ants? Please give a sample size of species in each category. I suggest uploading the compiled data you used in the analysis, since they are from two databases anyway. Response: Thank you. Seed dispersal is a complicated process and existing dispersal data is limited. Given this complexity, we tried our best to classify dispersal modes into autochory, endozoochory, exozoochory and anemochory according to the morphological features of their seeds or fruits. This approach provided consistency across our classification scheme, although we realize that in a few cases multiple modes of dispersal may be possible. The sample size for each dispersal category will be provided in the revised manuscript.

Third, you mentioned many important issues in analysing data, but you stop at only a brief mention. You need to continue with why they are important issues and how you have solved them. That is, you need to explain what this analysis does and why it is relevant to your data, rather than just referring to an R function. Since I could not find the function you mentioned in doing variation partitioning(L123), I doubt whether it can really be suitable to your data structure where you have a continuous variable (seed mass), categorical variables (dispersal mode, genus) and proportion data (seed mass variability) all mixed? Response: We will explain why the statistical methods are used to make the Methods section clear in the revised manuscript. Variation partitioning(VP) can be performed by 'rda' or 'varpart' functions in vegan package. VP is similar to regression analysis, which does not require the designation of the type of explanatory variable, and hence is suitable to our data structure.

In L122-123, it is good that you realise multicollinearity, but how did you treat variables with high collinearity issue? L126 – the "assumptions ofnormality" refers to the residuals, but have you checked your model residuals? L89-91:Why did you use this index to

show seed mass variability? Why not use (for example)coefficient of variation, which is more commonly used? Response: Our collinearity inspection, with variance inflation factor (VIF) <4, indicated that we didn't have variables with high collinearity issue in the models. We did not check the residuals of the statistical models. We appreciate the reminder! We used that index because it is more suitable than the coefficient of variation (CV). We had tried the CV to measure seed mass variability and obtained a similar result to what is shown in the manuscript, although the r-square value of the model - including the CV - was lower. The main disadvantage of the CV is that it becomes very sensitive to small variation of the mean when the latter is close to zero, and some plant species have very small seeds.

Fourth, there are some analytical flaws. L101 – 1 degree×1 degree grids cannot be of the same size across the globe. This is a wrong procedure in calculating species range size, because it seems you did not consider projection. Since your grids have different sizes near the equator and at high latitudes, how could the species range size be comparable across species? Response: We will include more detail and rewrite this section to make it more clear. Usually, we should transform geographic coordinates into projected coordinates. Here, we considered the projection and the range size was calculated by multiplying the number of grids by 12,388.

In addition, I doubt the use of genus to surrogate phylogeny. In the variation partitioning, it seems you did not incorporate the phylogenetic correlation among species, which violates your previous sentence "closely related species tend to have similar traits and interspecific analyses can be compromised by phylogenetic correlation". Response: We appreciate this comment. We were unable to find a variation partitioning model that could incorporate a phylogenetic tree. Therefore, we used genus as a surrogate in the phylogeny, which we agree is not a perfect method. In the variation partitioning, we didn't need to consider the phylogenetic correlation of the response variable (range size), since the phylogenetic tree (i.e., Genus) is included as an explanatory variable. We are however open to alternative suggestions for this analysis.

In L125 – are you saying you did ANOVA with post-hoc Tukey HSD tests? What package did you use to do the tests? Again, you need to take phylogeny into account, otherwise, residuals do not fit model assumptions. Response: We did the ANOVA using the 'glht' function in the multcomp package. We will take phylogeny into account in the revised manuscript, but we are not sure whether we can find a ANOVA model that can incorporate phylogenetic trees. We will explore this option further.

Minor comments: The Introduction is generally written but can be improved with clear predictions. You only present study goals and predictions until the last paragraph (L74-75 is not a clear hypothesis). Why you make such hypotheses and what makes your study novel are essential throughout the whole Introduction section. For example, in L39-40 "few studies have explored this relationship", then what have they found? What makes your study different from these previous ones? L50 – what do you plan to do about this gap? Response: Thank you for the above comments. These comments will significantly improve our manuscript. We will add the contents in the Introduction section accordingly.

L66: Rephrase "have the same time to dispersal". No idea what this means – this line either has a grammar error or is delivered wrongly. Response: We will rephrased this sentence.

All references do not have years – how could they be matched with citations in the main text? Response: That is a little odd. Publication years show properly in our version. In any case, we will follow-up on this in the revision.

Figure 1 and Figure 2 actually tell little information. I suggest removing Figure 2 since no tip can be seen with these many species. For Figure 1, I suggest using grids to show numbers of specimen records (same unit as the grids used in range size), which can avoid overlapping. Response: Thank you for your suggestions, which will be strongly followed.

Figure 4: Is range size log-transformed (Figure 4 and Table A2) or log10-

transformed(the main text)? What is the flat panel in each figure? Here, you standardised predictors, but this information is not given in the Methods section. Response: Range size was log10-transformed in Figure 4 and Table A2. Flat panel represented regression plane in Figure 4. We will add the standardized information in the revised manuscript for a further clarification.

Figure 5: Seed mass and seed mass variability are two separate variables, but why are they combined in the variation partition? The results in Figure 3 and Figure 5 are not reliable, due to the flawed methods. Response: Seed mass and seed mass variability represent the mean and the variance, respectively. They are two attributes of seed mass. In the revision, we will consider other more suitable models in Figure 3 and Figure 5.

Please also note the supplement to this comment:
https://bg.copernicus.org/preprints/bg-2020-186/bg-2020-186-AC2-supplement.zip

---

## Author Comment (AC3) · 21 Aug 2020

Thank you! Your comments have improved our manuscript.

―――――――――――――――